# Highly Reversible Zn Anodes through a Hydrophobic Interface Formed by Electrolyte Additive

**DOI:** 10.3390/nano13091547

**Published:** 2023-05-05

**Authors:** Xiaoying Yan, Yunwei Tong, Yingjie Liu, Xinyu Li, Zhenbo Qin, Zhong Wu, Wenbin Hu

**Affiliations:** 1School of Materials Science and Engineering, Tianjin Key Laboratory of Composite and Functional Materials, Key Laboratory of Advanced Ceramics and Machining Technology (Ministry of Education), Tianjin University, Tianjin 300072, China; yxy970819@tju.edu.cn (X.Y.); tongyunwei@tiju.edu.cn (Y.T.); yingjie_liu@tju.edu.cn (Y.L.); xinyu9925@gmail.com (X.L.); qinzhb@tju.edu.cn (Z.Q.); wbhu@tju.edu.cn (W.H.); 2Joint School of National University of Singapore and Tianjin University, International Campus of Tianjin University, Fuzhou 350207, China

**Keywords:** aqueous electrolytes, Zinc-ion batteries, electrolyte additive, HER, hydrophobicity

## Abstract

Hydrogen evolution reaction and dendrite growth seriously break the Zn plating/stripping process at the electrolyte/electrode interface, causing the instability of the Zn anode of aqueous zinc ion batteries. To improve the Zn anode stability and reversibility, we report a new electrolyte additive of aqueous electrolyte with the hydrophobic group. This interfacial hydrophobicity maximises the exclusion of free water from the Zn anode surface, which blocks water erosion and reduces interfacial side reactions. Thus, in an optimal 2 M ZnSO_4_ electrolyte with 2 g·L^−1^ Tween-85, the hydrogen evolution reaction and other water-induced undesired reactions can be suppressed, which greatly improves the cycling stability and Coulombic efficiency (CE) of Zn plating/stripping process. The stable cycle time of the Zn//Zn symmetric battery reaches over 1300 h, especially at a high current density and a high areal capacity (more than 650 h at 5 mA·cm^−2^, 5 mAh·cm^−2^). The average Coulomb efficiency (CE) of Zn//Ti asymmetric cell achieves 98.11% after 300 cycles. The capacity retention rate of Zn//MnO_2_ full battery is up to 88.6% after 1000 cycles.

## 1. Introduction

With the over-exploitation of non-renewable energy sources such as oil, coal, and natural gas, which produce greenhouse gases and toxic products causing serious environmental pollution and energy crisis problems, the development of clean and sustainable energy sources has received widespread attention [1,2,3]. Lithium-ion batteries have been used on a large scale in energy storage [4,5]. However, limited lithium resources, high prices and safety issues have prevented the large-scale adoption of lithium-ion batteries [6]. In contrast, aqueous zinc-ion batteries show good promise in several aspects, including cost and safety, and are expected to be used for large-scale energy storage [7,8,9,10].

In aqueous electrolytes, Zn^2+^ is usually hydrated with water molecules and desolvated during the deposition of Zn, which allows water molecules to enter the double electric layer of the Zn anode. The decoupled water molecules can be easily reduced on the surface of the Zn anode, as water is thermodynamically destabilized at the redox potential of Zn, which will lead to hydrogen evolution and corrosion [11,12,13,14]. In order to solve the undesired unreversible reaction between water and Zn anode, researchers have proposed methods such as surface hydrophobic coatings, in situ waterproof SEI films and organic electrolytes [15,16,17,18]. For example, the addition of DMSO [19] and DMC [14] to the electrolyte can form a water-resistant solid electrolyte interphase (SEI) layer in situ on the zinc anode surface; a hydrophobic polystyrene coating [20] on the zinc anode surface allows zinc ion migration, but repels water penetration from the electrolyte to the zinc surface. However, there are still limitations to the large-scale application of these methods. In situ SEI films suffer from low strength, poor flexibility and low ionic conductivity, resulting in poor zinc deposition/stripping efficiency and excessive polarization at high current densities. Artificial surface coatings suffer from a complex preparation process and lack the self-healing ability and poor cycling stability. Although the complete removal of aqueous solvents can be achieved by using organic electrolytes, there is a risk of combustion and explosion in such electrolyte systems.

Therefore, in order to maximize the exclusion of water from the zinc anode surface while reducing interfacial side reactions and maintaining fast reaction kinetics in aqueous electrolytes, hydrophobicity can be introduced directly into aqueous electrolytes by adding electrolyte additives with hydrophobic groups. In this work, Tween-85 (T-85) with three hydrophobic groups (CH_2_)_7_CH(C_8_H_17_)CH_3_ was chosen as an electrolyte additive which can be adsorbed at the anode/electrolyte interface via the -OH group in the ZnSO_4_ electrolyte system. Its hydrophobic groups can be close together to form a hydrophobic interface, which promotes the hydrophobic properties of the Zn surface to prevent water erosion [21]. A large number of hydrophobic groups forms a good barrier between the Zn anode and the electrolyte, preventing the contact of water molecules with the Zn anode and inhibiting the hydrogen evolution reaction.

In addition to Tween-85, the Tween range of surfactants also includes Tween-20, Tween-40, Tween-60 and Tween-80. The members of the Tween family have the same hydrophilic group, while the hydrophobic group varies. Tween-20 contains a C14 saturated alkyl chain, Tween-40 and Tween-60 both contain a C18 saturated alkyl chain, Tween-80 has a double bond in its C18 alkyl chain and Tween-85 contains three C18 unsaturated alkyl chains. Each Tween-85 molecule contains three alkyl chains, which allows it to produce the strongest hydrophobic aggregation. Therefore, we chose Tween-85 as the electrolyte additive to form the hydrophobic interface.

## 2. Experimental

### 2.1. Electrode and Battery Fabrication

Preparation of electrolyte. A 1000 mL volumetric flask was used to prepare 1000 mL of 2 mol·L^−1^ ZnSO_4_ electrolyte. Using a pipette gun, 100 mL of the configured ZnSO_4_ electrolyte was taken and then different levels of T-85 were added to configure control electrolytes containing T-85 at concentrations of 0.02 g·L^−1^, 0.2 g·L^−1^, 2 g·L^−1^ and 5 g·L^−1^.

Preparation of zinc electrodes. All zinc foils were sanded with 200, 800 and 2000 mesh sandpaper to remove the oxide layer before using and placed in the ethanol solution, ultrasonic for 5 min, and dried naturally in the air and cut into 14 mm diameter discs.

Preparation of α-MnO_2_. α-MnO_2_ was synthesized by hydrothermal method. A total of 0.003 M MnSO_4_·H_2_O was dissolved in 60 mL of deionised water, 2 mL of 0.5 M H_2_SO_4_ was added and the solution was stirred until clarified. Then, 0.002 M KMnO_4_ was dissolved in 20 mL deionised water and added drop by drop to the above solution after stirring well. After stirring for 3 h at room temperature, the solution was transferred to a reactor and reacted hydrothermally at 120 °C for 12 h. After three washes with water, three washes with alcohol and centrifugation, the sample was dried in an oven at 60 °C for 12 h to obtain the α-MnO_2_.

Preparation of positive electrode sheets. The α-MnO_2_ powder was first mixed with Acetylene black, polyvinylidene fluoride (PVDF) in a weight ratio of 7:2:1 using *N*-methyl-2-pyrrolidone (NMP) as a solvent. The mixed positive paste was then evenly coated onto 14 mm diameter carbon paper and dried at 60 °C for 24 h before use. The loading mass of α-MnO_2_ was 1.0 mg·cm^−2^.

Battery assembly. The battery was assembled in the order of positive shell, positive plate, diaphragm, electrolyte, negative plate, gasket, shrapnel andnegative shell and compacted with a tablet press. The battery was placed in the air for 8 h for subsequent testing.

### 2.2. Characterization

The crystal structures of Zn anodes were gathered by X-ray diffractometry (XRD, Bruker D8 Advanced, X’Pert, Cu Kα, Karlsruhe, Germany). The morphology of the Zn anode was obtained by scanning electron microscopy (SEM, TESCAN MIRA LMS, Tokyo, Japan). The contact angles between different electrolytes and Zn electrodes were measured by using the contact angle meter (Data physics OCA20, Stuttgart, Germany).

### 2.3. Electrochemical Tests

The Neware Battery Test System (BTS-5V 50 mA) was used to carry out tests related to zinc ion batteries. The long cycling performances of Zn//Zn symmetric cells were performed under 1 and 5 mA·cm^−2^ current density with 1 and 5 mAh·cm^−2^, respectively. The Coulombic efficiency of the plating and stripping process was measured by using asymmetrical Zn//Ti cells under 1 mA·cm^−2^ current density. For Zn//MnO_2_ batteries, the long cycling performance was tested at a current density of 1 C (1 C = 308 mA·g^−1^).

All electrochemical tests were carried out using the electrochemical workstation (CHI 760E). The Tafel curves of Zn anode were measured using a three-electrode system, in which zinc foil played as the working electrode, Pt worked as the counter electrode and the Ag/AgCl electrode acted as the reference electrode. The hydrogen evolution reaction (HER) curve was tested directly with Zn//Zn cells, using a two-electrode system. Cyclic Voltammetry (CV) was tested directly with Zn//MnO_2_ cells using a two-electrode system.

## 3. Results and Discussion

### 3.1. Electrochemical Characterizations of the Zn Plating/Stripping Performance

The ionic conductivity is an important parameter to measure the nature of the electrolyte and characterizes the resistance to ionic conduction. The ionic conductivity of the ZnSO_4_ electrolyte with different levels of T-85 was measured using a conductivity meter and the results are shown in Figure 1a. The ionic conductivity was 49.6, 50.3, 51.6, 48.2 and 45.4 mS·cm^−1^ at concentrations of 0, 0.02, 0.2, 2 and 5 g·L^−1^ of T-85 in the ZnSO_4_ electrolyte, respectively. A small increase in the ionic conductivity of the electrolyte was observed as the concentration of T-85 increased, while at a concentration of 2 g·L^−1^ the conductivity decreased slightly and was lower than that of the pure ZnSO_4_ electrolyte, and the conductivity decreased further at the concentration of 5 g·L^−1^. This indicates that the addition of excess T-85 caused a greater resistance to the transfer of anions and cations in the electrolyte, which can also be referred to as polarization. Further, the effect of different levels of T-85 electrolyte on the stability of the Zn plating/stripping cycle was investigated by means of Zn//Zn symmetric cells. Constant current charge/discharge tests were carried out at a current density of 1 mA·cm^−2^ (1 mAh·cm^−2^), as shown in Figure 1b. The detailed results are as follows, (1) the cycle time of the Zn//Zn symmetric cell was 100 h without the use of T-85, (2) the cycle time of the symmetric cell was extended to 150 h at the T-85 concentration of 0.02 g·L^−1^, (3) the cycle time of the symmetric cell was further extended to 750 h at the T-85 concentration of 0.2 g·L^−1^, (4) the cycle time of the symmetric cell reached a maximum of 1300 h at the T-85 concentration of 2 g·L^−1^, (5) at a concentration of 5 g·L^−1^ of T-85, the symmetric cell cycle time decreased to 530 h. In short, as the concentration of T-85 increased, the cycle lifespan of the symmetric cells appeared to be extreme, which indicates that the low T-85 content does not have a significant improvement on the cycle lifespan of the Zn anode, but when the T-85 content is too high, the polarization voltage is larger, and the cycle lifespan no longer increases further. At a T-85 concentration of 2 g·L^−1^, the Zn anode achieves the longest cycle lifespan of over 1300 h. Therefore, considering the ionic conductivity of the electrolyte and the changing pattern of electrochemical stability of the Zn//Zn symmetric cells, a ZnSO_4_ electrolyte with a T-85 content of 2 g·L^−1^ was chosen for subsequent studies.

Long-cycle tests with Zn//Zn symmetric cells have shown that in the system without T-85, the symmetric cell was short-circuited after only 100 h of cycling, probably due to the growth of zinc dendrites piercing the glass fibre diaphragm and short-circuiting inside the cell, leading to cell failure. In contrast, symmetric cells with T-85 can cycle stably for over 1300 h at a current density of 1 mA·cm^−2^ without any short circuit or failure. This can demonstrate that the addition of T-85 can significantly improve the stability of the zinc plating/stripping cycling process. However, for aqueous zinc ion batteries, the poor performance at high current densities and large discharge area capacities is a limitation. Therefore, we increased the current density of the symmetrical battery to 5 mA·cm^−2^ and increased the discharge area capacity to 5 mAh·cm^−2^. The results of the long-cycle test are shown in Figure 2. After 100 h of cycling at a current density of 5 mA·cm^−2^, the symmetric cell without T-85 suddenly developed a short circuit. The symmetric cell with T-85, on the other hand, despite having a larger polarization voltage (95 mV), could still cycle steadily for 450 h, more than six times the cycle life of the cell without T-85. This indicates that T-85 maintains the cycling stability of the Zn anode at different current densities.

Comparing the T-85 additive modification strategy with the artificial coating method or in situ SEI method, which also has a hydrophobic effect, this method avoids the problems of poor cyclic stability of the artificial coating and the low strength and flexibility of the in situ SEI membrane. Therefore, compared to other related studies in recent years [20,22,23,24,25,26,27,28,29], the present work shows great advantages, as shown in Figure 3. The cumulative capacity is at a relatively high level of 1300 mAh·cm^−2^ at a current density of 1 mA·cm^−2^. The advantage of higher current densities (5 mA·cm^−2^) is more obvious with a high cumulative capacity (2250 mAh·cm^−2^).

Coulomb efficiency (CE) is one of the most important parameters for evaluating the reversibility of zinc plating/stripping [30]. As shown in Figure 4a, in the Zn//Ti asymmetric cell without T-85, the average CE was only 94.32% for the first 70 cycles and declined rapidly to 38% after 70 cycles. This is due to the formation of “dead zinc” and by-products due to the occurrence of dendrites and side reactions during the cycling process, resulting in a CE decrease. However, after the addition of T-85, the CE was significantly improved and remained stable even after 300 cycles, with an average CE of 98.11%, indicating that T-85 improved the by-products and the “dead zinc” problem and improved the reversibility of zinc plating/striping. Furthermore, the voltage hysteresis is the value of the potential difference that occurs when Zn^2+^ is embedding /detaching from the positive and negative materials, indicating a large resistance to the conduction process of Zn^2+^ in the electrolyte. According to the previous discussion, 2 g·L^−1^ of T-85 causes a decrease in the ionic conductivity of the electrolyte, which implies an increase in the ohmic impedance of the electrolyte and, therefore, an increase in the polarization voltage. Figure 4b shows the voltage distribution for different electrolyte systems. With T-85, the polarization voltage is 118 mV, which is higher than the polarization voltage of cells without T-85 (48 mV). Therefore, the use of T-85 leads to an increase in the overpotential during zinc plating/stripping, but the small increase in polarization voltage does not have a significant impact on the stability and reversibility of Zn anode plating/stripping.

### 3.2. Modification of the Hydrogen Evolution Reaction in Zn Anodes

The molecular structure of T-85 is shown in Figure 5a. In the ZnSO_4_ electrolyte system, its hydrophilic part CHO(CH_2_CH_2_O)_n_CH_2_CH_2_OH is adsorbed at the anode/electrolyte interface through the hydrogen bond formed between the -OH group therein and the water molecules. On the other hand, the main hydrophobic part (CH_2_)_7_CH(C_8_H_17_)CH_3_ extends into the electrolyte. In this way, the large number of hydrophobic groups forms a good barrier between the Zn anode and the electrolyte, preventing water molecules from coming into contact with the Zn anode and inhibiting hydrogen evolution reactions from occurring. The contact angles of the different electrolytes on the Zn foil surface were measured by means of a contact angle measuring instrument, as shown in Figure 5b. The contact angle between the electrolyte without T-85 and the Zn foil was 84°, while the contact angle between the electrolyte with T-85 and the Zn foil was 92°, thus verifying the hydrophobicity of T-85.

Hydrogen precipitation is an unavoidable side reaction in aqueous electrolyte solutions, which has the following negative effects on the battery, (1) hydrogen precipitation requires partial charge consumption during the charging process and low Coulomb efficiency, (2) excessive power is wasted, (3) the generation of hydrogen is accompanied by the production of OH^-^, which increases the local pH at the electrode/electrolyte interface and exacerbates the production of the by-product (ZnSO_4_[Zn(OH)_2_]_3_·12H_2_O), (4) hydrogen production causes the zinc layer to produce more holes, tips and dendrites, etc., (5) hydrogen precipitation will increase the internal pressure of the sealed cell, causing it to swell [31]. Based on the above issues, inhibition of the hydrogen evolution reaction is important to enhance the stability of the Zn anode. As shown in the HER curve (Figure 6a), the hydrogen evolution overpotential was −0.048 V (vs. Zn/Zn^2+^) in the system without T-85, whereas with T-85 the hydrogen evolution overpotential shifted significantly negatively to −0.13 V (using the potential corresponding to a current density of 10 mA·cm^−2^ as the standard), indicating a positive effect of T-85 on the inhibition of the hydrogen evolution reaction. Corrosion reaction is another major problem with Zn anode, which was explored using Tafel curves (Figure 6b). The corrosion potential represents the probability of the corrosion reaction occurring at the Zn anode, the corrosion current is the current corresponding to the corrosion potential and is used as a measure of the rate at which the corrosion reaction occurs at the Zn anode [32]. The increase in corrosion potential with the addition of T-85, compared to the system without T-85, indicates that the Zn anode has a reduced tendency to corrode in the presence of T-85. The corrosion current is also reduced compared to the system without T-85, indicating that T-85 also contributes to a reduction in the corrosion reaction rate. The reduction in the probability and rate of corrosion reactions indicates that T-85 effectively inhibits the corrosion reaction of the Zn anode and contributes to the reversibility of Zn plating/stripping. 

In order to observe the corrosion phenomenon of zinc anodes in the ZnSO_4_ electrolyte more visually, the surface morphology and structure of the zinc foil were characterized by immersing it in the ZnSO_4_ electrolyte containing different concentrations of T-85 for 7 days. In the ZnSO_4_ solution, without the addition of T-85 (Figure 7a), severe corrosion occurred on the surface of the zinc foil, with an uneven surface and many deep corrosion pits, while the accumulation of insoluble corrosion products could be observed. In contrast, in the ZnSO_4_ solution with 0.02 g·L^−1^ and 0.2 g·L^−1^ T-85 (Figure 7b,c), the corrosion of the zinc foil decreased, and the surface became flatter. When the concentration of T-85 was increased to 2 g·L^−1^ (Figure 7d), the surface of the zinc foil was smoother, and no significant corrosion was observed. From the XRD results (Figure 7e), the surface of the zinc foil in the ZnSO_4_ solution without T-85 had a clear and intense by-product peak, which gradually disappeared as the concentration of T-85 increased. Therefore, based on the SEM and XRD results, it can be seen that T-85 can inhibit the corrosion of Zn anodes in ZnSO_4_ electrolyte, improve the surface morphology of zinc and inhibit the generation of by-products, and the higher the concentration, the more obvious the inhibition effect.

In addition, the changes in the surface of the Zn anode after cycling of the Zn//Zn symmetric cells in the presence/absence of T-85 were explored, using SEM and XRD characterisation of the Zn anode surface after 50 cycles at 5 mA·cm^−2^, 5 mAh·cm^−2^. The surface of the Zn anode of the symmetric cell without T-85 was rough and uneven after 50 cycles, accompanied by visible corrosion pits (Figure 8a). This indicates that more severe corrosion has occurred, and that the corrosion reaction has exacerbated the non-uniformity of zinc deposition, with protrusions having entered the glass fibre diaphragm, whereas the addition of T-85 resulted in only some very small pits on the surface and a smoother overall surface (Figure 8b). This indicates that T-85 inhibits corrosion well during the cycling of the symmetric cells and inhibits dendrite growth successfully. Figure 8c shows the XRD pattern corresponding to each of the Zn anodes after 50 cycles in different electrolytes. The main by-product of the weakly acidic electrolyte is the alkaline zinc sulphate Zn_4_SO_4_(OH)_6_·5H_2_O (ZHS). It is produced as a result of a chemical reaction between the electrolyte and the OH^-^ produced by the hydrogen evolution reaction. During charging, large amounts of Zn^2+^ are near the surface of the negative electrode. At the same time, hydrogen evolution leads to an increase in OH^-^ concentration near the surface, which leads to the formation of ZHS. The Zn anode without T-85 showed distinct peaks at 2θ = 8.2°, 16.3° and 24.5°, implying the formation of the by-product ZHS [33,34]. In contrast, no corresponding peaks appeared for the Zn anode using T-85. This indicates that there is almost no by-product generation during the cycle after the addition of T-85, which can effectively inhibit the formation of by-products. The insoluble and non-conductive by-products ZHS attached to the electrode surface consume Zn^2+^ and additional electrons, resulting in a low Coulomb efficiency (CE). In addition, the by-products cover the active sites for zinc plating/stripping and increase the surface inhomogeneity, which accelerates the growth of dendrites. Therefore, in combination with the HER curves, the XRD results and the surface morphology of the zinc anode shown in the SEM pictures, the addition of T-85 not only inhibits the hydrogen evolution reaction and suppresses the formation of by-products, but also mitigates the problem of dendrite growth.

### 3.3. Electrochemical Performance of the Full Batteries

To further test the practicality of the T-85 additive in zinc ion batteries, Zn//MnO_2_ full batteries were assembled for electrochemical performance testing. 0.2 M MnSO_4_ is added to the electrolyte to inhibit the dissolution of Mn^2+^ from the positive electrode of MnO_2_. The negative electrode is zinc foil, and the positive electrode is α-MnO_2_ synthesized by the hydrothermal method. The hydrothermal method was used to synthesise α-MnO_2_, which is a tunneling structure with good stability and can maintain a stable structure during Zn^2+^ embedding/detaching. Figure 9a shows the SEM image of α-MnO_2_, which has the morphology of a nanorod-like structure. Figure 9b shows the XRD spectrum of α-MnO_2_, which can correspond well with the standard crystalline phase of α-MnO_2_ (PDF#44-0141) [35], with no impurity peaks observed. Based on the SEM and XRD results, it indicates that α-MnO_2_ was successfully synthesized.

Figure 10a shows the cyclic voltametric curves of the Zn//MnO_2_ full batteries in different electrolyte systems, showing two pairs of redox peaks corresponding to the two-step reaction process. The oxidation peaks in the interval of 1.5 V to 1.7 V represent the Zn^2+^-removal process, and the reduction peaks in the interval of 1.2~1.3 V and 1.3~1.4 V represent the Zn^2+^-embedding process. The addition of T-85 increased the polarisation in the full battery and hindered the ion transport, which is consistent with the previous studies. The full battery using T-85 had a higher redox peak current compared to the unused full battery, implying an increase in battery capacity. Further, the Zn//MnO_2_ full batteries were tested for long-cycle stability at 1C current density (1C = 308 mA·g^−1^ based on MnO_2_), as shown in Figure 10b. The full battery with T-85 maintained 88.6% specific capacity after 1000 cycles, whereas the battery without T-85 showed a greater specific capacity decay over 1000 cycles, with a capacity retention of only 45%. This suggests that T-85 improves the long-cycle stability of the full battery. From the multiplier performance comparison in Figure 10c, the battery discharge capacity was increased to some extent with the addition of T-85 at different current densities and remained stable throughout the increase in current density. Therefore, the use of T-85 also has a degree of improvement in the full battery multiplier performance.

The self-discharge problem of the full battery is related to the self-corrosion of the Zn anode in the battery, which occurs when the battery is in the resting state and affects the capacity of the full battery. Therefore, a full battery self-discharge test was carried out and used to investigate the self-corrosion of the Zn anode in the full battery in the resting state. In order to eliminate instability in the initial charge/discharge works, the full batteries of the two different electrolyte systems were first cycled five times at a current density of 1 C, then charged to 1.8 V and left to stand for 48 h before continuing to discharge to 1 V, and the test results obtained are shown in Figure 10d. The voltage of the full battery without T-85 dropped to 1.45 V after 48 h of standing, while the voltage of the battery with T-85 only dropped to 1.48 V. This suggests that the addition of T-85 inhibited the self-corrosion behavior of the full battery to some extent.

## 4. Conclusions

A common and low-cost electrolyte additive Tween-85 with hydrophobic groups was introduced into the 2 M ZnSO_4_ electrolyte. In the electrolyte, the hydrophobic groups can be closely oriented to each other to form a hydrophobic interface which contributes to the hydrophobic properties of Zn anode surface to block the erosion of water. As a result, water-induced undesired reactions are suppressed, which can improve the cyclability and reversibility of the Zn anode. With the addition of Tween-85, the Zn anode exhibits excellent cycle durability (over 1300 h) and a high coulomb efficiency of 98.11% for Zn plating/stripping on titanium foil (300 cycles). The Zn//MnO_2_ full battery achieves 88.6% capacity retention after 1000 stable cycles at a current density of 1 C.

## Figures and Tables

**Figure 1 nanomaterials-13-01547-f001:**
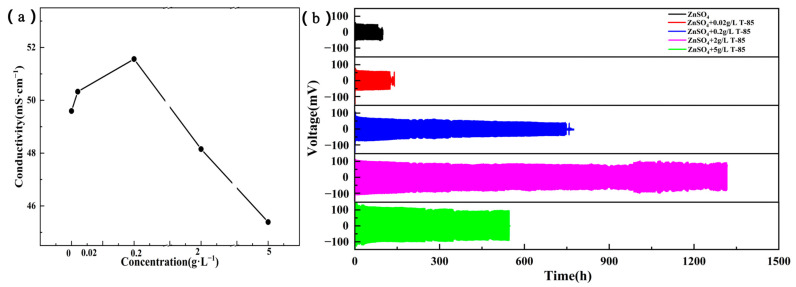
(**a**) Ionic conductivity of different electrolyte systems; (**b**) Constant current charge/discharge curves of Zn//Zn symmetric cells at 1 mA·cm^−2^, 1 mAh·cm^−2^.

**Figure 2 nanomaterials-13-01547-f002:**
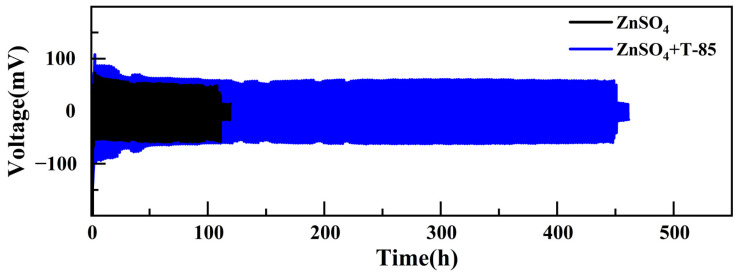
Constant current charge/discharge curves for Zn//Zn symmetric cells at 5 mA·cm^−2^, 5 mAh·cm^−2^.

**Figure 3 nanomaterials-13-01547-f003:**
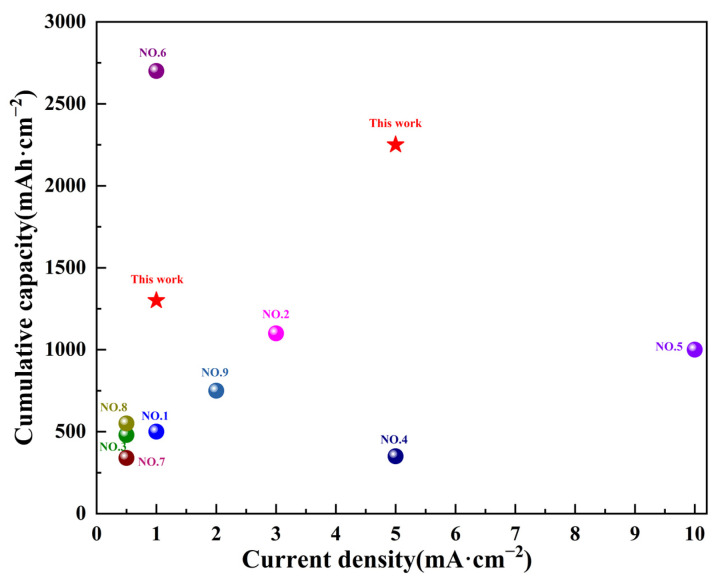
Comparison of current density and cumulative capacity of symmetric cells using a hydrophobic effect modification strategy.

**Figure 4 nanomaterials-13-01547-f004:**
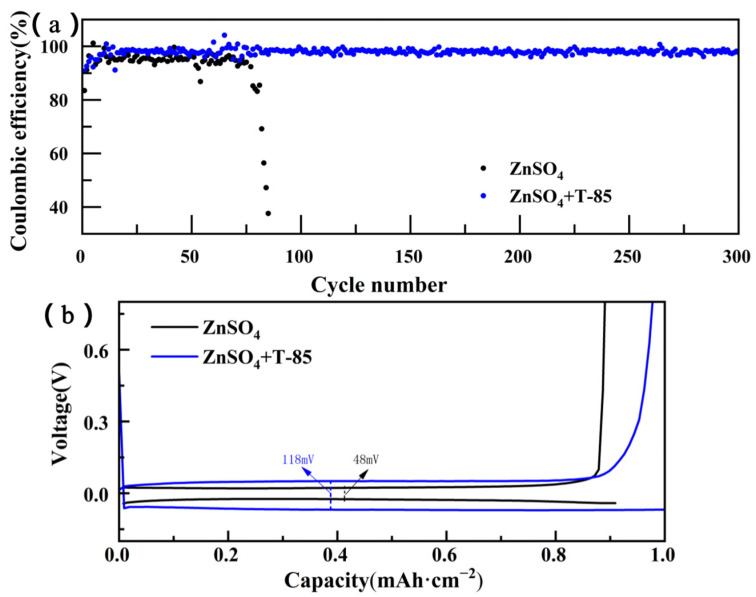
(**a**) Zn plating/stripping Coulomb efficiency graph at 1 mA·cm^−2^, 1 mAh·cm^−2^; (**b**) Voltage distribution curve of Zn//Ti cells with different electrolyte systems.

**Figure 5 nanomaterials-13-01547-f005:**
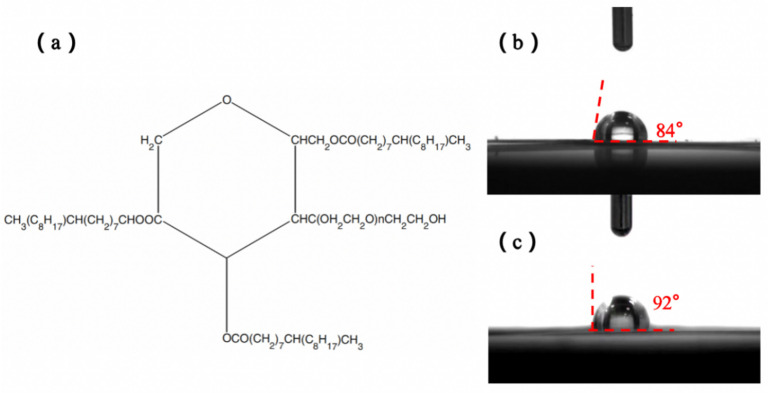
(**a**) Molecular structure of Tween-85; Contact angle of different electrolyte systems on the surface of Zn foil (**b**) without Tween-85, (**c**) with Tween-85.

**Figure 6 nanomaterials-13-01547-f006:**
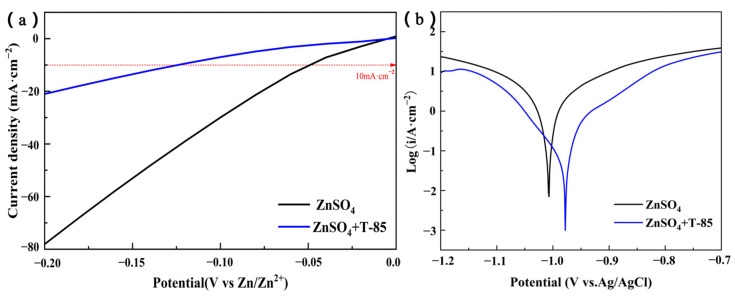
(**a**) HER curves and (**b**) Tafel curves for Zn anodes in different electrolyte systems.

**Figure 7 nanomaterials-13-01547-f007:**
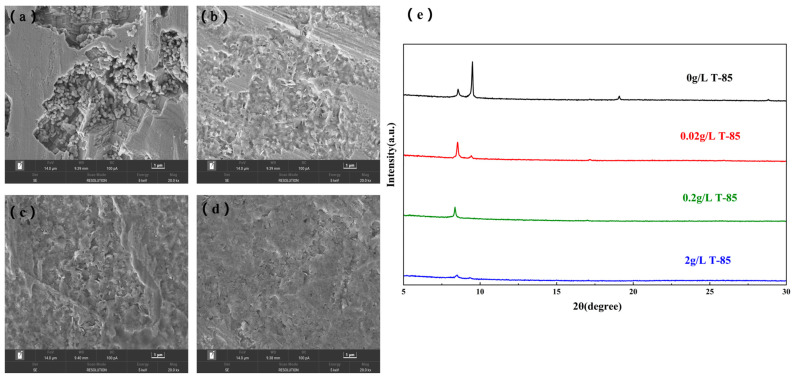
Zn foil in ZnSO_4_ solutions with different levels of T-85 after 7 days of resting. SEM images (**a**) 0 g·L^−1^; (**b**) 0.02 g·L^−1^; (**c**) 0.2 g·L^−1^ (**d**) 2 g·L^−1^; (**e**) XRD pattern.

**Figure 8 nanomaterials-13-01547-f008:**
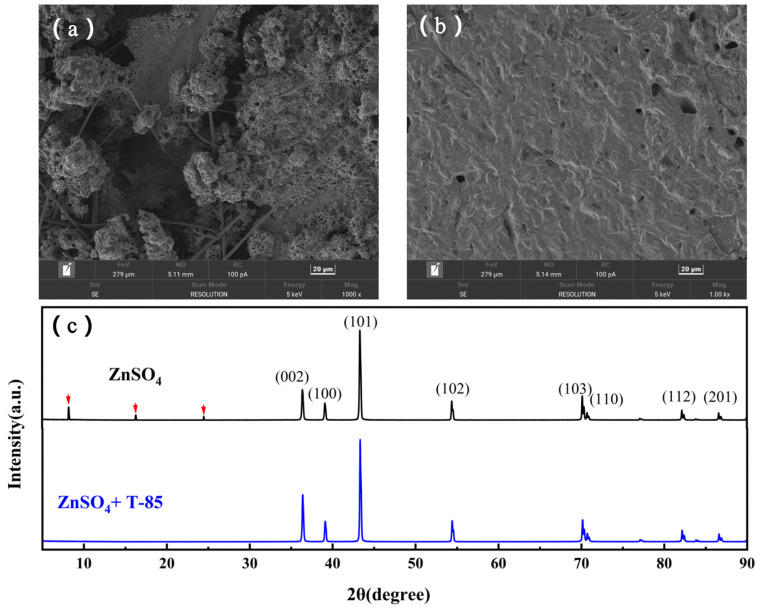
(**a**,**b**) SEM image and (**c**) XRD pattern of Zn//Zn symmetric cells with Zn anode after 50 cycles.

**Figure 9 nanomaterials-13-01547-f009:**
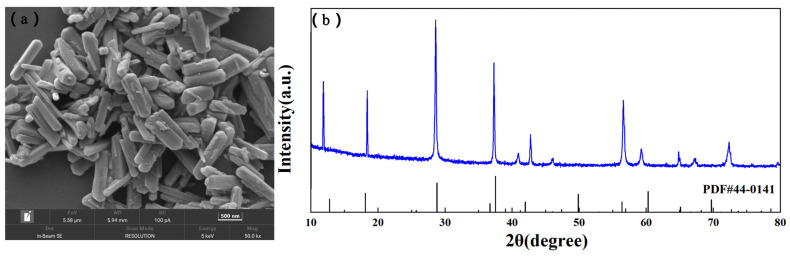
(**a**) SEM image and (**b**) XRD pattern of α-MnO_2_.

**Figure 10 nanomaterials-13-01547-f010:**
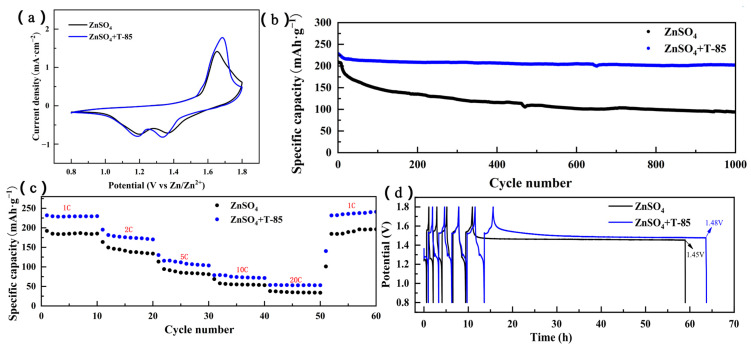
(**a**) Zn//MnO_2_ full cells with different electrolyte systems (**a**) Cyclic voltametric curve; (**b**) Long cycle performance at 1C current density; (**c**) Multiplier performance; (**d**) Self-discharge test.

## Data Availability

Not applicable.

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
