# Peer review of "Highly Reversible Zn Anodes through a Hydrophobic Interface Formed by Electrolyte Additive"

_nanomaterials, 2023, doi:10.3390/nano13091547_

Round 1
Reviewer 1 Report
1. In line 4, “with hydrophobicity” is inappropriate because the additive cannot be dissolved in water if it is hydrophobic dominant. Tween-85 has an HLB value of 11, leading to a microemulsion if its concentration is higher than the CMC of Tween-85.
2. In line 16, “less active water” is not a proper concept because the bulk water cannot be activated.
3. In line 36, solventise should be hydrated.
4. In lines 60-62, as the dosages of T-85 used (line 70) are all higher than CMC (0.06 mM) of this detergent, both micelles and vesicles should adsorb on the Zn anode rather than what is portrayed in line 61. The dosages of T-85 are higher than its critical micelle concentration.
5. The authors have to add a description to the Last paragraph of the Introduction for why T-85 but not the other Tween family members were selected.
6. The presence of T-85 micelles in the electrolyte will decrease the activity of Zn2+ ions due to their adsorption on the micelles.
Reviewer 2 Report
The manuscript, ‘Highly reversible Zn anodes through a hydrophobic interface 2 formed by electrolyte additive’ demonstrate a method to suppress HER and Zinc dendrite growth effectively by using Tween-85 non-ionic surfactant as an additive with variable concentrations ranging from 0.02 to 5g·L-1 in the aqueous 2M ZnSO4 electrolyte. The study is reasonably designed and well-conducted. I recommend it for publication only after the minor revision, and also the authors have to address the important issues/comments mentioned below;
1. Why does the polarization voltage increases to 118 mV using T-85 compared to 48 mV and not have any significant impact on the reversibility and stability during Zn plating/stripping?
2. Assign the XRD peaks with their miller indices values in Figure 8b.
3. A comparison table in the supporting information, comparing the cycling performances of the symmetric cells between this work and other reported works will improve the quality of the work and its readability.
4. There are some typos, check lines 45, 46 (check again whether it is cathode or anode), and 328 (it should be Figure 9d).
5. Unify all the references according to the Journal’s style in the reference section.
A minor spell check is required throughout the manuscript.
